# Human-hippo conflicts around Lake Tana Biosphere Reserve, Ethiopia: Vulnerability of hippopotamus in human-dominated landscape

**Zewdu Kifle** [ORCID]*, **Workiyie Worie Assefa, Amera Moges**

Bahir Dar University, College of Science, Bahir Dar, Ethiopia

* zewdu96@yahoo.com

## Abstract

Currently, the demand of the human population for more land, water, and other natural resources from wildlife habitats is increasing all over the world. Such intense human pressure results in conflict with wildlife and the impacts affect both parties negatively. The human-hippo conflict poses a serious problem for both local farmers' livelihoods and hippo conservation. To date, the extent of human-hippo conflict is poorly documented in Ethiopia. Specifically, the extent of human-hippo conflicts around Lake Tana Biosphere Reserve (LTBR) is unknown. Therefore, this study aimed to investigate the extent of human–hippo conflict, and possible mitigation measures proposed by the local people around LTBR, Ethiopia. We conducted a questionnaire interview with the household head, the household head's wife, or other adults $\geq$ 18 years old. All respondents reported that crop damage was the main cause of human–hippo conflict around LTBR. Livestock grazing competition (17.4%) and human attack (19.5%) were also sources of conflicts in the region. Respondents claimed that hippos destroyed crops including maize (*Zea mays*), teff (*Eragrostis teff*), finger millet (*Eleusine coracana*), and rice (*Oryza sativa*). Most (91.2%) respondents claimed that the severity of crop damage caused by hippos was high in the region. Most respondents (range 90 to 93%) complained about high crop damage suggesting that hippos be eliminated from the region. Local people estimated that the population sizes of hippos comprise an average of 243 individuals; however, we counted 122 hippos during our boat survey in the area. The result of this study showed that human-hippo conflicts cause negative effects on both farmers' livelihood and hippo conservation in the region. To mitigate human-hippo conflict, we suggest that proper land use zonation systems around key areas, broad awareness creation among local people, and alternative crop production should be promoted around the LTBR.

## Introduction

Human population growth and the demand for more land, water, and other natural resources occupy wildlife habitats through agricultural expansion, habitat destruction, and encroachment. These bring overlap between people and wildlife over the resources that lead to human-

**Data Availability Statement:** All relevant data are within the paper.

**Funding:** We all the authors (Zewdu Kifle, Workiyie Worie Assefa and Amera Moges) received the

award. Unfortunately, the award has no grant number. The funder of the research work was Bahir Dar University. URL of Bahir Dar University website https://www.bdu.edu.et/ No.

**Competing interests:** The authors have declared that no competing interests exist.

wildlife conflict [1–4]. When conflict is intensified it hurts both humans and wildlife [4, 5]. Wild animals are hurt through habitat loss, retributive killing in revenge to crop foraging and livestock predation, and illegal hunting for bushmeat by local people [6]. In turn, crop foraging and livestock predation by wild animals cause a significant economic impact on rural people's livelihoods [4, 7]. Thus, human-wildlife conflict is one of the leading conservation issues and concerns around the globe [2, 3, 8, 9], and it is as urgent as climate change [10]. Conflict often occurs when human settlements and agricultural lands are adjacent to wildlife habitats. For example, subsistence farmers grow crops and graze their livestock nearby lakes and rivers. People also have many interests to utilize the resources associated with wildlife-occupied wetland habitats which complicate conservation issues.

African communities living alongside large mammals incur the costs of conservation through crop damage and livestock loss but may gain little benefit through ecotourism [4, 11, 12]. The financial costs of human-wildlife conflict imposed on rural people can be very high [11, 13]. Moreover, the impacts of human-wildlife conflict cause indirect effects including the opportunity costs like time loss for guarding crops, sleeping loss, missing schools or underperform at school, and human fatality on the rural communities [11, 14–17].

Hippopotamus (*Hippopotamus amphibious*) (hereafter hippo) is an extant African megaherbivore. Hippo is large-bodied and has a huge mouth in which the upper and lower canine teeth and incisors are enlarged into tusks. The eyes and ears are small and are set far back and high on the head. The body is an elongated barrel shape, carried on short, stocky legs. Each foot has four toes, with thick nails [18]. They are long-lived (lifespan 50 years in captivity) animals with sexual maturity typically attained between seven and 15 years and with well-developed senses [18]. Hippos have small home ranges (mean of ~8 km$^2$) relative to other large herbivores [19]. They mainly feed on graminoids (grasses and sedges) [18]. Hippo crops graminoid entirely using its leathery lips leaving a short, smooth lawn. They usually feed within 2–3 km of water body boundaries [20], and they live in school up to 30 individuals usually six or fewer [18], and are territorial [18]. Hippos are found in rivers, lakes and permanently flooded wetlands, where individuals may remain submerged for up to six minutes [21]. They use swamps and waters as daytime refuges to keep cool, protect their skin from sunburn and avoid biting insects [20]. Hippo sweat contains a red and an orange pigment that acts as sunscreen [20].

Like other megaherbivores, hippos have a great influence on their environment [22]. Hippos play a great role to transfer and facilitate nutrients from terrestrial to aquatic ecosystems, through their excrements [23]. These nutrient subsidies influence the primary and secondary aquatic production and can determine the aquatic community composition [24, 25]. In addition, the grazing activity of hippos facilitates graminoids (grasses and sedges) to become short and are known as hippo lawns, in turn attracting a diverse grazing herbivore assemblage and enhancing spatial vegetation heterogeneity [26, 27]. Moreover, hippos play a significant role in fluvial geomorphology and nutrient transfer benefits to fish populations [28]. Hippos themselves provide a direct ecosystem service, as their meat may consume by people who hunted the animal legally or illegally [29, 30], the monetary gains derived through photographic tourism [31], sport hunting [32], and the medicinal values used by traditional healers [33, 34].

Hippo is affected by human activities and is one of the vulnerable mammals with declining population size and distribution across its range [35]. Hippos are vulnerable to extirpation, as body mass is a known extinction correlate among vertebrates [36]. The population of hippos is slow to respond to high rates of offtake, and may quickly become extirpated [36]. They have been reduced to a fraction of their former range through retributive killing [6], They are regionally extinct in five African countries (Algeria, Egypt, Eritrea, Liberia, and Mauritania) and near to extermination in Congo, Gambia, and Somalia and it is currently categorized as

vulnerable by the International Union for Conservation of Nature (IUCN) Red List [35, 37]. The present Anthropocene activities (e.g., habitat loss due to crop cultivation and overhunting for their meat, hides, and ivory) are the main threats to hippos [35, 37–39].

Megaherbivore mammals (with an adult body mass greater than 1,000 kg [20, 40] like hippos and elephants (*Loxodonta africana*) are ranked among the most problematic and lie at the heart of human-wildlife conflicts as they need wide areas concerning their size [40]. Although these megaherbivores often cause major devastation to crops and are often a physical threat to humans, most research has focused on the elephant and neglected the hippopotamus, yet the latter are involved in numerous conflicts with people in many parts of Africa [39, 41, 42]. Thus, the hippo is relatively understudied by scientists, and very little is known about human–hippo conflict, and experimental research on mitigation methods has hardly been conducted in the field [43].

Wetlands have a great biological significance in terms of harboring biodiversity and many vulnerable wildlife species like hippos need conservation focus there. Hippo is one of the wetland mammals of Ethiopia that frequently spends time in and at the edge of water bodies. Although human-wildlife conflict is a day-to-day occurrence in Ethiopia, few studies have been conducted and these studies were mainly focused on primates [4, 11, 16]. For that matter, the extent of the conflict between humans and hippos is poorly documented in the country [44]. In particular, there has been no systematic study on the extent of human-hippo conflict around Lake Tana (the largest lake in Ethiopia), Ethiopia. Lake Tana forms large areas of wetlands in its surroundings. These wetlands provide a myriad of goods and services for humans and animals [45]. However, such wetlands are converted into farmlands that create conflicts between local farmers and hippos. Therefore, the objectives of this study were to assess the cause of conflicts, examine the extent of the human-hippo conflict, and explore the attitudes and conservation interests of local people towards hippos in the region. Achieving these objectives is relevant for informing reserve managers for planning the conservation management and mitigation strategies by focusing on more sustainable approaches concerning human–hippo interaction in the region.

## Materials and methods

### Ethical statement

The review board of college of science of the Bahir Dar University (BDU) reviewed and approved the research proposal (S1 File). The research and community service vice president office review board reapproved and accepted the proposal. The questionnaire (S1 File) and its consent acquisition procedure were also approved by BDU. After human ethics research approval, we obtained permission letter that contained a letter of requesting support to all Kebeles and villages to conduct this research project around Lake Tana Biosphere Reserve from BDU.

At the beginning of each potential questionnaire interview, the aim of the research was briefly explained by the interviewers for each respondent. The interviewers also fully informed to the respondents on how the data obtained from them would be used. Confidentiality of information was assured by the investigators. Participants gave their consent orally after being explained the aim of the survey, topics of different sections on the questionnaire, and how the interview would proceed. Consent was obtained by having participants state that they agree to participate. Respondents were selected opportunistically based on chance encounter from each village. We undertook the interviews in the Amharic language since the residents of the area are Amhara community. In the meantime, we didn't access any medical issues in this research project in the area.

## Study area

This study was conducted around Lake Tana Biosphere Reserve (LTBR). The biosphere reserve comprises Lake Tana, the largest lake in Ethiopia (Fig 1). It is located between 11˚.50' to 12˚ 40' N latitude and 36˚ 90' to 37˚ 30' E longitude with an average elevation of 1800 m asl; the lake is the natural type which covers 309,132.12 ha with an average depth of 8 m [45]. Lake Tana is the main source of the Abbay River. The climate is characterized by a unimodal rainfall pattern, the wet season includes from June to October with peak rainfall in August, and the dry season is from November to May when decreased the water level of the lake region. The mean maximum and minimum temperatures of Lake Tana are 29.20 and 10.90 ˚C, respectively [46]. The Lake Tana area consists of more than 37 islands and 16 peninsulas, giving a home to 21 churches and monasteries dating back to the 13th century with a unique cultural, and religious heritage, and historical and aesthetic value.

The area is characterized by an enormous heterogeneity of land uses human settlements, villages, farmlands, livestock grazing pastures, and natural ecosystems. Many wetlands are located almost all around Lake Tana and are ecologically the most important units in Ethiopia. The reserve is a hotspot of biodiversity, internationally known as an Important Bird Area. The LTBR is an important fish resource and is home to up to 28 different species of

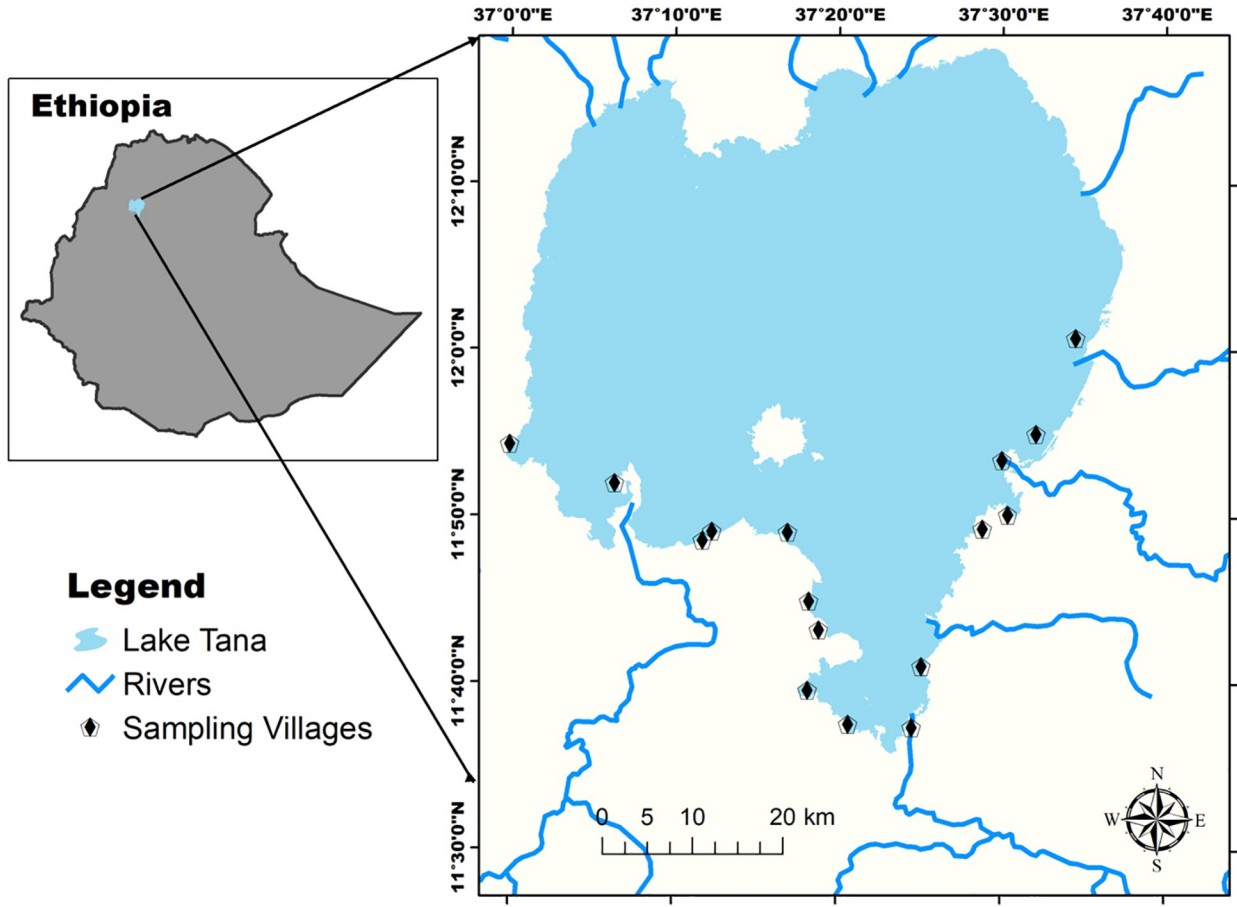

**Fig 1. Location Lake Tana Biosphere Reserve, Ethiopia.** (the map has been created by our team using shapefiles acquired from the Ethiopian Mapping Agency. These shapefiles can be accessed for free through the official website of the agency at http://www.ema.gov.et/).

fish of which 20 species are endemic to the Lake Tana catchment [47]. LTBR is part of the Eastern Afromontane Biodiversity Hotspot. The main economic activities of the region are agriculture, fishing, and national and international tourism (religious and recreational). The area has a unique cultural, historical, and aesthetic value with numerous monasteries and churches. Farming expands to wetlands including the shorelines of the lake and at some pocket areas when the lake level drops during the dry season. Several kilometers of dry parts of the lake area become available for agriculture and are used by the local farmers to grow cereal crops [48].

Common grass species of Family Poaceae of the area include *Eragrostis* spp., *Pennisetum* spp., *Panicum* spp., *Echinochloa* spp., *Setaria* spp., *Phragmites australis*, *Hyparrhenia* spp., *Cymbopogon* spp., and *Sorghum* spp. In addition, Family Acanthaceae (*Hygrophila auriculata*), Family Cyperaceae (*Cyprus papyrus*), Family Typhaceae (*Typha latifolia*), Nymphaeaceae (*Nymphaea caerulea*), Family Juncaceae (*Juncus dregeanus*), Family Commelinaceae (*Floscopa glomerata*), Family Eriocaulaceae (*Eriocaulon* spp.), and Family Xyridaceae (*Xyris capensis*) are the main species of wetlands of the region (personal observation). LTBR and its surrounding areas also support a diverse mammalian community including hippos (*Hippopotamus amphibious*), olive baboons (*Papio anubis*), grivet monkeys (*Chlorocebus aethiops*), bushpigs (*Potamochoerus larvatus*), rock hyraxes (*Procavia capensis*), African clawless otters (*Aonyx capensis*), Nile monitors (*Varanus niloticus*) and different bird and fish species as well as invertebrate animals (personal observation).

## Data collection

We conducted questionnaire interview surveys of households in 15 villages surrounding the LTBR. Only household heads aged 18 years and above were eligible for inclusion in this questionnaire interview survey. We included only one participant from the household/family living within the same house to recruit respondents. We conducted the questionnaire survey in the three parts of the LTBR (South, East, and West) from Sep ten to Nov 30, 2013. The total sample sizes of respondents who were participated on this questionnaire interview survey were 204 individuals. We didn't perform the sample size and power calculation before participants were recruited due to difficulties (e.g., time and logistics) to obtain the actual total population size inhabiting around LTBR. However, we assumed these participants (204 individuals) can represent the total human population live along the boundary of LTBR. We recruited respondents opportunistically based on a chance encounter adopted by [4, 16, 49] from each village.

Using a questionnaire, we collected information on the socioeconomic and demographic character of the respondents (age, sex, education level, household size, and livestock size), cause and extent of conflicts, crop types damaged by hippos, time of crop damage, human and livestock mortality and injury, and the attitudes and conservation views of respondents towards hippos, number of hippos around their villages, and possible suggestions as conflict mitigation measures.

In addition, we performed a direct count of hippos using a boat survey. During the boat survey, the research team observed hippos by driving around the lakeshore in the early morning and late afternoon. When we observed hippos, the researchers halted the boat for 15–30 minutes and counted the number of individuals from far using naked eyes and/or binoculars. We observed and recorded groups of hippos at several pocket sites associated with wetlands and in the shallow parts of the lake where more vegetation cover occurs. We observed the local farmers watching their cropland from hippos, especially during the dawn and dusk when hippos attempt to come out of the lake for feeding.

## Data analysis

We presented the data collected from the interviews as the percentage frequency of respondents giving each response in the case of multi-response questions. We used descriptive statistics to examine patterns in the human–hippo conflict including conflict type, damaged crop, and the solution to the conflict.

# Results

## Socioeconomic background of the respondents

In this questionnaire survey, we interviewed 204 respondents comprising 96.1% males and 3.9% females (Table 1). Respondents' age ranged from 18 to 75, with a mean of 38.2. Most (97.1%) respondents had never been in school at all. All respondents depended on farming and livestock keeping as the main source of livelihood. Some (18.1%) respondents supported their livelihood using fishing activities.

## Complaints of local farmers against hippos

All respondents complained that they had encountered conflict with hippos as a result of cereal crop damage (Table 2). In addition to crop damage, these respondents claimed that aggression against human beings (19.5%) was the other form of conflict followed by competition with livestock (17.4%) in the region. In addition, many respondents reported that hippos damaged fishermen's fishnets. As a result of these, most respondents considered hippos as their enemies.

**Table 1. Summary of socioeconomic and demographic profile of the respondents around LTBR.**

| Characteristics | Frequency | Percentage (%) |
|---|---|---|
| Sex | | |
| Male | 196 | 96.1 |
| Female | 8 | 3.9 |
| Age groups (years) | | |
| <31 | 75 | 36.8 |
| 31–50 | 92 | 45.1 |
| >50 | 37 | 18.1 |
| Education level | | |
| No education | 156 | 76.2 |
| Primary school | 41 | 20.3 |
| Religion | 7 | 3.5 |
| Livelihood source | | |
| Crop production and livestock | 167 | 81.9 |
| Fishing | 37 | 18.1 |
| Household size | | |
| Five person or less | 110 | 53.9 |
| Six person and above | 94 | 46.1 |
| Livestock size | | |
| <5 | 83 | 40.7 |
| 5–10 | 76 | 37.1 |
| >10 | 45 | 22.2 |

**Table 2. Cause of human-hippo conflict found around LTBR.**

| Form of conflict | Frequency | Percentage (%) |
|---|---|---|
| Crop damage | 293 | 100.0 |
| Grazing competition with livestock | 51 | 17.4 |
| Livestock killing/injure/biting | 25 | 8.5 |
| Human death/injure/aggression | 57 | 19.5 |

## Crop types damaged by hippos

Cereal crop species damaged by hippos around LTBR were maize (*Zea mays*), finger millet (*Eleusine coracana*), teff (*Eragrostis tef*), and rice (*Oryza sativa*). These crops are predominantly cultivated in the area. Most (91.0%) respondents stated that crop feeding predominantly occurred when crops were at the vegetative stage. Few (5.6%) respondents reported that crop feeding occurred at the seedling stage. The other (2.4%) respondents reported that the crop damage occurred during the maturation stage while the other (1.0%) respondents reported that the damages occurred at all growing stages. Local farmers reported that crop damage by hippos happened not only by eating but also through their broad feet prints during walking on standing crops of the farmlands.

## Crop damage period and extent by hippos

All respondents reported that crop damage attempts by hippos exclusively occurred during night hours. None of the respondents reported daylight crop damage by hippos in the region. Most respondents stated that they looked after their crop fields throughout the night hours from being damaged by hippos. Respondents claimed that the extent of crop damage is high in the region. Most (91.2%) respondents claimed that the extent of crop damage caused by hippos is severe, medium (7.8%), and insignificant (1.0%) in the region. Most (90.7%) respondents claimed that hippos made their livelihood to be complicated through crop damage, livestock injury, sleep loss, and biting by mosquitoes and other insects during guarding cereal crops at night. Some of these respondents hated hippos, and they considered them devils. The other (9.3%) respondents did not consider those problems as serious. In addition, most respondents reported that crop damage by hippos increased (89.9%); others (11.1%) reported that crop damage is stable year after year. Most respondents reported that recession farming intensified the extent of human-hippo conflicts in the region.

## Awareness of hippo population size and conservation status

When asked about the population size of hippos, an average of 243 populations were inhibited in the east, south, and west part of LTBR (Table 3). However, we directly counted 122 hippos during our survey on those mentioned sites. When we asked about the status of the hippo population size, most (93.6%) interviewees reported that the population size increased, 5.4% believed that the number of hippos remained constant and 1.0% believed they have decreased in their number in recent years.

## Attitudes of the local people toward hippos

Most (89.2%) respondents reported that they had a negative attitude toward hippos. The other (10.8%) respondents reported that they had a positive attitude toward hippos. Those respondents who had a positive attitude towards hippos associated this animal with God and the tourism sector. Most (83.8%) interviewees did not appreciate the presence of hippos in their

**Table 3. Population estimation of hippos in the surveyed area; SD: Standard deviation.**

| Village | GPS Point | | Survey Direction | Number of respondents | Respondent's mean estimate | SD | Our survey estimate |
|---|---|---|---|---|---|---|---|
| | Latitude | Longitude | | | | | |
| kunzela_Estumit | 282173 | 1316767 | West | 10 | 40 | 18.7 | 10 |
| Abbay Ras | 293783 | 1312404 | West | 41 | 65.5 | 34.1 | 36 |
| Lijome | 303492 | 1306035 | West | 19 | 19.5 | 33.1 | 11 |
| Sekelet | 304546 | 1307053 | South | 24 | 5.5 | 1.9 | 4 |
| Ambo Bahir | 315282 | 1299299 | South | 6 | 5 | 0.6 | 4 |
| Robet_Koreta | 327765 | 1292033 | East | 18 | 8.5 | 8.6 | 13 |
| Mitsele_Gugube | 337344 | 1308796 | East | 4 | 12 | 8.4 | 3 |
| Kirstose Semira | 336721 | 1314835 | East | 31 | 36 | 19 | 22 |
| Nabega_Teza Amba | 344902 | 1328350 | East | 21 | 13 | 8.1 | 9 |
| Abagerima | 319602 | 1285641 | South | 5 | 6 | 2.3 | Not seen |
| Wejeta | 315136 | 1289436 | South | 6 | 2.2 | 2.0 | Not seen |
| Monzi | 316379 | 1296092 | South | 5 | 4 | 3.1 | Not seen |
| Atista | 312942 | 1306947 | South | 4 | 6 | 4.1 | Not seen |
| Abbay Mewucha | 326655 | 1285285 | South | 6 | 16 | 5.5 | 7 |
| Rema Medihenialem | 334534 | 1307259 | East | 4 | 4 | 3.8 | Not seen |
| Sendeye | 340500 | 1317728 | East | - | - | - | 3 |
| **Total** | | | | **204** | **243** | | **122** |

localities. Most (95.7%) respondents claimed that hippos have no ecological as well as economic benefit; instead, the respondents claimed that hippos are affecting their livelihood. Most (89.9%) of these local farmers reported that they wanted to eliminate hippos from their localities. Respondents reported that hippos are frequently killed by the local farmers in their localities as revenge of crop damages and livestock causalities. We observed two freshly dead hippos which were killed by local farmers during our boat survey periods around lakeshore farmlands. The killings happened through automatic gun firings. These killing confirmations of hippos by the local people showed that there is a highly intense degree of human-hippo conflicts in the region.

## Reactions of people against crop damage

The local people used different crop-guarding methods to protect their crops from being damaged by hippos (Table 4). Among the respondents, 38.7% of the local farmers used a combination of a gun firing, whip cracking, or horn to protect their crops from hippo damage. Most respondents reported that they stayed in the watchtower to guard their crop fields throughout the night.

**Table 4. Crop protection methods used by local community.**

| Protection methods | Frequency | Percentage |
|---|---|---|
| Torch/firelight | 49 | 24.0 |
| Sling/stoning/spearing | 42 | 20.6 |
| Dung or droppings smoking/benzene spraying | 17 | 8.3 |
| Whip cracking/gun firing/horn/noise | 79 | 38.7 |
| Fencing using thorny twigs | 17 | 8.3 |

## Possible mitigation measures

All respondents expected mitigation measures from government officials. Most (93.7%) respondents claimed that they would be happy if hippos are eliminated from their localities. Few (6.3%) respondents reported that hippos should be transferred to the lake sites where crop cultivations are not practiced for their safety.

## Discussion

Hippopotamus is a megaherbivore mammal that occupies wetlands associated with lakes and rivers [21], and each adult hippo may consume 4050 kg (wet mass) of forage daily [23]. To date populations of the hippopotamus have undergone a decline as a result of habitat conversion and loss, anthropogenic factors such as hunting/killing due to resource sharing, and conflicts with humans [50, 51]. Human-wildlife conflicts are intensified in human-dominated landscapes, with significant implications for species conservation worldwide [4, 8, 11, 16]. Like other developing countries, Ethiopia has experienced human-wildlife conflict due to the exponential human population growth that requires more land for settlement and cultivation [4, 11, 16]. Wetlands are threatened by the conversion into farming land and overgrazing by livestock as a result hippos expand their range towards the farmland for resource acquisition. Even though habitat degradation and wetland loss due to human population growth are the primary drivers of human-wildlife conflicts, only a few studies have been conducted to evaluate the extent of conflicts in the country. Specifically, sociological studies on the human-hippo conflict have not been carried out in Ethiopia. Thus, the extent of conflicts between humans and hippos remains largely unstudied in the country. Therefore, this study was conducted to examine the extent of human-hippo conflict around LTBR, Ethiopia. Understanding conflicts between the local farmers and hippos is crucial for the development of sustainable conservation and management strategies around LTBR.

Human-hippo conflict principally occurs when people practice subsistence-level farming and fishing [52]. Crop damage is the most common human-hippo conflict in LTBR. Similarly, crop damage was the most common type of conflict between humans and hippos in Kenya [40]. This might be due to the occurrence of high agricultural activities and the dense human population in the region. Similar studies showed that farming shifted towards wetlands including river banks and the shoreline of the lake within the last 2 decades, due to an increase in population pressure [53]. Such cultivation trends towards wetlands and river banks could have a great impact on the environment and wildlife conservation. In recent decades, rice cultivation has also become the major crop in floodplains of the lake [53] where frequent conflicts occurred between farmers and the hippo. Similarly, conflicts are high in regions where hippo coexists with high human population densities, particularly close to major wetlands and rivers [40]. In addition, the proximity of hippo habitats near livestock grazing and farming aggravates the conflicts near LTBR. This pattern is consistent with the observation that human–hippo conflicts are linked to increasing human population and the associated increase in demand for agricultural and settlement space, especially in areas close to water, and the fact that hippos, humans, and their livestock compete for resources along wetland margins [40]. Subsistence farming near water sources may motivate people to farm, thereby exacerbating conflict [54, 55]. The closer the crops are to water bodies, the higher the likelihood of being damaged by hippos [56, 57]. In addition, hippos are competitors with livestock over grasses and sedges to access the same foraging areas [26]. The depletion of natural forage by livestock may also drive hippos to feed crops, which aggravates conflict with people [57].

From our results, the main livelihood activity of the local population around the LTBR was farming. This study found that around LTBR, crop damage by hippos had a significant

negative impact on the livelihood of local farmers and reduced local tolerance for the species. Most respondents reported that maize and teff are destroyed by hippos in the region. Maize, finger millet, and teff are the most widely grown crops around LTBR. Maize, rice, sorghum, cowpeas, pumpkins, groundnuts, sweet potatoes, cassava, sugarcane, cabbage, and are crops damaged by hippos [57, 58]. Additionally, most respondents around LTBR stated that crop feeding by hippos predominantly occurred at the vegetative stage. Crop feeding by hippos may be affected by crop growth stages [56, 57].

Our results showed that local people developed a negative attitude toward hippos as a result of crop damage, livestock injury, sleep loss, and biting by insects during guarding crops at night. These impacts may complicate the livelihood of the local farmers, and they develop negative perceptions towards hippos. Similarly, crop damaged by mega mammals like elephants lead to a negative perception of the species among local people across Africa [59, 60]. Crop damage by elephants hurt the local social economy and leads to reduce local tolerance for the species around Moukalaba-Doudou National Park in Gabon [7]. Crop damage is the main reason for the negative perceptions of local communities on hippos [29, 57, 61].

Although we did not explicitly quantify the economic loss in this study, hippos have caused severe crop losses, require night hours in the field just to guard crops, and can have a substantial effect on the local farmer's livelihood. For example, a study conducted in Namibia in 2009 indicated that crop damage by hippos was estimated at 2,193 USD per hectare [32]. Crop feeding by hippos occurs mostly at night [56]. Similarly, in this study, most respondents stated that hippos can cause crop damage during nighttime and that, once they enter a crop field, they cause widespread damage through feeding and trampling. Similarly, hippos damaged crops by trampling [56]. Such crop damage may increase negative attitudes toward hippos regardless of the actual amount of loss. Crop damage caused by elephants around Moukalaba-Doudou National Park in Gabon is one of the major threats to the local population as it is detrimental to livelihood, which depends largely on agriculture [7]. Similarly, a study conducted in Namibia showed that the costs of crop damage were greater than the income generated by hippos through tourism and hunting [32].

Farmers around LTBR used various methods (e.g., torch lighting, slinging, dung or benzene smoking, fencing using thorns, whip cracking, and gun firing) to protect their crops from hippo damage. Most respondents used guarding methods using slinging to protect their cereal crops from being damaged by hippos. Likewise, acoustic deterrents like shouting and the banging of drums or bells [57], light-based deterrents like the lighting of bonfires and torches [62], and physical barriers like thorn bush bomas and sisal (*Agave sisalana*) fences [63] and stone walls [57] are used as crop protection methods against hippos in different African countries. But individuals should be available near their farmlands throughout the night hours. Thus, farmers' crop protection methods are labor-intensive, but these are the only affordable options [64]. In addition, an increase in agricultural labor to prevent crop damage depending on manpower, such as guarding the fields overnight, has taken a toll on locals' physical and mental health [15, 65–67]. Children may also terminate school, and adults may miss their wives at night to guard crops. These long-term impacts and burdens may force the local people to kill hippos in mass.

## Conclusion and conservation implications

Habitat loss through unsustainable agricultural practices and habitat degradation are the main threats to wildlife in Ethiopia. Wetlands have great biological significance in terms of harboring biodiversity and are important sites for biodiversity conservation in Ethiopia. However; they are regarded as vulnerable zones, and some are on the verge of total damage due to over-

utilization and urbanization [68]. These extensive conversions of natural habitats (e.g., wet-lands) into agricultural lands have significant impacts on both local people and wildlife as the result of human-wildlife conflicts. This study provides crucial information on the extent of human-hippo conflict around LTBR. The results indicate that hippos caused significant crop losses in the region. Crop damage, in particular, has deprived people of not only food resources but also opportunity costs like sleeping loss and biting by insects at night. All these bring local people to develop negative attitudes towards hippos and their conservation. In the absence of effective measures against human-hippo conflicts, people's attitudes towards this mammal highly deteriorated over time.

The major threats to the hippopotamus around the study area are habitat degradation through the expansion of agriculture, competition with livestock, and killing by the local people due to crop damage. As the human population and food demand increase, the human-hippos conflict is likely to become more serious and worsen with a projected increase in anthropogenic impacts in the region. Moreover, as the human populations continue to grow; human-hippo conflict will be a leading driver of hippo extirpation in the region. This paper provides useful information to conservation decision-makers and managers and highlights some possible conservation measures. Thus, we recommend establishing a buffer zone to minimize recession farming. Instead, local farmers should grow cash crops including mango, avocado, papaya, orange, and lemon near hippo habitats to minimize the conflicts. The conservation plan should reduce the human impact on the grazing land of hippos by creating alternative job opportunities (e.g., fish pond farming and livestock fattening) by the government for local farmers and youths to vivid economic benefit. These alternative job opportunities can minimize the human demand and load on the hippo habitats. In addition, the conservation plan should include community educational awareness programs and community-based participation by coordinating local and district zone responsible offices. These education and outreach programs should focus on the cultural value of hippos to the community as well as the role of hippos as ecosystem engineers and tourist attractions that support local economies. We also recommend conducting hippo population surveys and human-hippo observation at night and the spatial aspect of conflicts to get a full picture of the population size and conflict in the region. Finally, we suggest undertaking long-term ecological and behavioral research on hippos, exploring the role of hippos in disease transfer to cattle, conducting experimental trials on how hippos respond to deterrent methods to implement the best effective mitigation approaches, and quantifying the nature of anthropogenic-related threats to hippos and their habitats in more detail in the region.

## Supporting information

**S1 File. Questionnaire and associated variables that we used for this study.**
(DOCX)

## Acknowledgments

We want to thank Bahir Dar University for its financial and logistic support. We thank the Amhara National Regional State Environment and Forest Protection Authority for permitting this research project in the region. We are also very thankful to the local people who participated in this study for questionnaire interviews. We thank Christopher Dutton (the reviewer) and another anonymous reviewer for their valuable comments and suggestions that helped us to improve the manuscript.

## Author Contributions

**Conceptualization:** Zewdu Kifle.

**Data curation:** Zewdu Kifle, Workiyie Worie Assefa, Amera Moges.

**Formal analysis:** Zewdu Kifle, Workiyie Worie Assefa, Amera Moges.

**Funding acquisition:** Zewdu Kifle.

**Investigation:** Zewdu Kifle, Workiyie Worie Assefa.

**Methodology:** Zewdu Kifle, Amera Moges.

**Project administration:** Zewdu Kifle.

**Resources:** Zewdu Kifle, Workiyie Worie Assefa, Amera Moges.

**Software:** Zewdu Kifle.

**Supervision:** Zewdu Kifle, Workiyie Worie Assefa.

**Validation:** Zewdu Kifle, Workiyie Worie Assefa, Amera Moges.

**Visualization:** Zewdu Kifle.

**Writing – original draft:** Zewdu Kifle.

**Writing – review & editing:** Zewdu Kifle, Workiyie Worie Assefa, Amera Moges.

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
