## [Decision Letter · Decision Letter 0]

13 Jun 2023

PONE-D-23-07127Human-Hippo Conflicts around Lake Tana Biosphere Reserve, Ethiopia: Vulnerability of Hippopotamus in Human-Dominated LandscapePLOS ONE

Dear Dr. Kifle,

Thank you for submitting your manuscript to PLOS ONE. After careful consideration, we feel that it has merit but does not fully meet PLOS ONE’s publication criteria as it currently stands. Therefore, we invite you to submit a revised version of the manuscript that addresses the points raised during the review process.

We look forward to receiving your revised manuscript.

Kind regards,

Frank O. Masese, Ph.D

Academic Editor

PLOS ONE

Journal Requirements:

Reviewers' comments:

Reviewer's Responses to Questions

**Comments to the Author**

1. Is the manuscript technically sound, and do the data support the conclusions?

Reviewer #1: Yes

Reviewer #2: Yes

2. Has the statistical analysis been performed appropriately and rigorously? 

Reviewer #1: Yes

Reviewer #2: Yes

3. Have the authors made all data underlying the findings in their manuscript fully available?

Reviewer #1: Yes

Reviewer #2: Yes

4. Is the manuscript presented in an intelligible fashion and written in standard English?

Reviewer #1: Yes

Reviewer #2: Yes

5. Review Comments to the Author

Reviewer #1: the paper is scientifically sound and supported by data that was obtained through questionnaires. all the relevant statistical data was presented and analysis of the statistics made. the paper has an abstract that outlines the argument, methods of research and findings which are key aspects. the introduction is well written and gives a clearly explained context of the study. methods and materials follow in a step by step approach that list and discuss each of the variables. then a discussion of the data is given with clear details and arguments. after the discussion comes the recommendations and conclusions.

Fore more information please refer to the uploaded reviewer's comments.

Reviewer #2: This article is on human-hippo conflict in an understudied region of Ethiopia. The authors did a preliminary hippo survey to count the hippos near different villages and then used questionnaires to understand how the locals in each of the villages respond to human-hippo conflict.

The authors found that there is significant human-hippo conflict and that most locals would prefer the hippo were not there. They found that hippos were responsible for significant crop damage. The authors recommend a proper land use zonation system around the LTBR, more awareness programs for the locals and potentially alternative crop production schemes.

I enjoyed reading the manuscript. I have a few suggestions that I hope will help improve it.

See attached Reviewer Response file for my line by line comments.

6. PLOS authors have the option to publish the peer review history of their article (what does this mean?). If published, this will include your full peer review and any attached files.

Reviewer #1: No

Reviewer #2: **Yes: **Christopher Dutton

---

## [Author Response · Author response to Decision Letter 0]

31 Jul 2023

July 10 2023

 Ethiopia 

Ref.: PONE-D-23-07127

Dear Frank O. Masese, Ph.D,   

This is a revision of our manuscript entitled “Human-Hippo Conflicts around Lake Tana Biosphere Reserve, Ethiopia: Vulnerability of Hippopotamus in Human-Dominated Landscape "by Zewdu Kifle, Workiyie Worie Assefa, and Amera Moges. On 13 Jun 2023, I received revision request in order to reconsider for publication. You invited me to submit a revised version of the manuscript that addresses the points raised during the review process. Hence, I addressed all the points and comments raised by the reviewers’ within this revised version of the manuscript. 

Below there are lists of the revisions. 

Sincerely,

Zewdu Kifle, Ph.D,

Correspondence Author 

Response for Editor and reviewers 

Thank you for your critical comments. I highlighted revisions with green color. In addition, I highlighted with red color and strikethrough for those deleted letters, words, phrases or sentences. 

Thank you!

Response to the Editor

1. Okay! We think that our manuscript meets PLOS ONE's style requirements. 

2. Well! All relevant data are available within the paper. 

3. Thank you for your comment on Figure 1. We presented our prepared map in this resubmitted manuscript. 

4. Okay! We reviewed our reference list. It is complete and correct. 

Response for reviewers

Reviewer #1

General comments: Thank you for appreciating the manuscript and its importance to understand human-hippo conflict in the region. We corrected those ideas based on your interesting comments. 

Specific comments:

1. Line 12: Okay! W changed it.

2. Line 28: Okay! We recommended to do further survey to determine accurate population size of hippos in the region.

3. Line 39-40: Well! Thank you! 

4. Line 41-43: Thank you! We restructured the sentence. 

5. Line 48: We changed it.

6. Line 52: Well! We restructured the sentence.

7. Line 58: Well! We added the citation.

8. Line 59: Okay! Thank you! We changed it. 

9. Line 61: We added the citation here.

10. Line 66: Thanks! We changed it.

11. Line 66-68: Okay! We added this paper. 

12. Line 71: Okay! We added a citation here. 

13. Line 79: Okay! Thank you!

14. Line 137: Well! We changed it.

15. Line 186: We removed the digit.

16. Table 3: Thank you! We added number of respondents and Standard deviation (SD) for each village. 

17. Line 218: Okay, We changed it.

18. Line 220: Okay, We changed it.

19. Line 226: Well, we changed it.

20. Line 227: Okay, We added it.

21. Line 231: Okay, We changed it.

22. Line 242: Okay! We corrected it. 

23. Line 244: Thank you! We corrected this sentence.

24. Line 256: Okay! We did it. 

25. Line 264: We changed it.

26. Line 267: Okay, Thanks!

27. Line 284: We changed this word.

28. Line 288: Well! We deleted it.

29. Line 300: We changed it. 

Reviewer #2

General comments: Thank you for admiring the manuscript and its offers in the area of human-wildlife conflict. We added more information based on your interesting comments to strengthen the paper.

Specific comments:

Abstract: 

1. Okay, W summarized the findings based on your comment. 

Introduction:

1. Line 39-40: Okay! We corrected it. 

2. Lines 1-51: We wrote this part in more detailed.

Materials and Methods:

1. Okay! We pointed key points on the map. 

2. Line 121: Okay! We mentioned villages involved in data collection section, and the numbers of respondents here. 

3. Line 124: Chance encounter avoids biases. We cited about it in the text. This method is very important to collect data from randomly encountered respondents within a short period of time and it is also unbiased tool. 

4. Line 136-37: Okay! We corrected this part. We changed ‘become out’ to ‘come out’.

5. Line 145-46: We have no particular reason. But in Ethiopian culture females give priorities for their husband. Thus, when we got both, we interviewed the males than the females. In addition, we encountered males in the field nearby the lakeshore in most cases too. Thus, we had no any particular reason why female respondents were 3.9% other than this. 

6. Line 155-56: We didn’t ask about human provocation as a cause of hippo aggression.

7. Line 172-73: Okay! Thank you, you right, the majority of hippos would be out at night. In this paper, we recommend to conduct hippo population survey at night and to get a full picture of the conflict in the region. Thus, such recommendation is very helpful for our paper by providing future population survey and human-hippo conflict observation directions. Thus, our survey doesn’t affect the paper.

8. Line 192 –Table: During our interview questionnaire surveys, we asked the respondents to estimate number of hippo populations inhabiting around their villages. Since, local people frequently are encountered with hippos around their farmlands; they are good sources to estimate hippos around their villages. We asked managers (wildlife management authorities) of the number of hippos in the LTBR. But, they didn’t have data about the population size (number) of hippos in the area. Thus, we couldn’t compare our gathered hippo statistics with the data of managers of the area. Our survey is the first in the area. 

9. Line 201-202: There is no a difference between local farmers and residents of the area. But to avoid confusion, we deleted residents and replaced by the local farmers. 

10. Line 202-205: you see, we observed two freshly dead hippos in our single survey. If we follow such killings in year round or more, we can get more dead hippos. We ascertained the killings by asking the local people. They informed us that the local farmers killed them through automatic gun firing. 

Discussion:

1. Line 239: All natural resources should be conserved. Thus, such cultivation trends towards wetlands and river banks could have a great impact to the environment and wildlife conservation. 

The managers of LTBR know the problem (human-hippos conflict) in the area. But, they do not have any scientific data to incorporate in this paper or discussion parts. 

2. Line 291: The government should create alternative job opportunities to minimize human demand and load on the hippo habitats.

---

## [Decision Letter · Decision Letter 1]

6 Sep 2023

Human-Hippo Conflicts around Lake Tana Biosphere Reserve, Ethiopia: Vulnerability of Hippopotamus in Human-Dominated Landscape

PONE-D-23-07127R1

Dear Dr. Kifle,

We’re pleased to inform you that your manuscript has been judged scientifically suitable for publication and will be formally accepted for publication once it meets all outstanding technical requirements.

Kind regards,

Frank O. Masese, Ph.D

Academic Editor

PLOS ONE

Additional Editor Comments (optional):

Reviewers' comments:

Reviewer's Responses to Questions

**Comments to the Author**

1. If the authors have adequately addressed your comments raised in a previous round of review and you feel that this manuscript is now acceptable for publication, you may indicate that here to bypass the “Comments to the Author” section, enter your conflict of interest statement in the “Confidential to Editor” section, and submit your "Accept" recommendation.

Reviewer #1: All comments have been addressed

2. Is the manuscript technically sound, and do the data support the conclusions?

Reviewer #1: Yes

3. Has the statistical analysis been performed appropriately and rigorously? 

Reviewer #1: Yes

4. Have the authors made all data underlying the findings in their manuscript fully available?

Reviewer #1: Yes

5. Is the manuscript presented in an intelligible fashion and written in standard English?

Reviewer #1: Yes

6. Review Comments to the Author

Reviewer #1: The revised paper is now grounded and can now be published. The authors have addressed the comments that were raised. They have provided the data and statistics which makes it easier for the reader to follow their analysis and argument.

7. PLOS authors have the option to publish the peer review history of their article (what does this mean?). If published, this will include your full peer review and any attached files.

Reviewer #1: No

---

## [Editor Report · Acceptance letter]

27 Sep 2023

PONE-D-23-07127R1 

Human-Hippo Conflicts around Lake Tana Biosphere Reserve, Ethiopia: Vulnerability of Hippopotamus in Human-Dominated Landscape 

Dear Dr. Kifle:

I'm pleased to inform you that your manuscript has been deemed suitable for publication in PLOS ONE. Congratulations! Your manuscript is now with our production department. 

Kind regards, 

on behalf of

Dr. Frank O. Masese 

Academic Editor

PLOS ONE